# A New Defense Against Adversarial Images: Turning a Weakness into a Strength

Tao Yu[*][†]    Shengyuan Hu[*][†]    Chuan Guo[†]    Wei-Lun Chao[‡]    Kilian Q. Weinberger[†]

## Abstract

Natural images are virtually surrounded by low-density misclassified regions that can be efficiently discovered by gradient-guided search — enabling the generation of adversarial images. While many techniques for detecting these attacks have been proposed, they are easily bypassed when the adversary has full knowledge of the detection mechanism and adapts the attack strategy accordingly. In this paper, we adopt a novel perspective and regard the omnipresence of adversarial perturbations as a strength rather than a weakness. We postulate that if an image has been tampered with, these adversarial directions either become harder to find with gradient methods or have substantially higher density than for natural images. We develop a practical test for this signature characteristic to successfully detect adversarial attacks, achieving unprecedented accuracy under the white-box setting where the adversary is given full knowledge of our detection mechanism.

## 1    Introduction

The advance of deep neural networks has led to natural questions regarding its robustness to both natural and malicious change in the test input. For the latter scenario, the seminal work of Biggio et al. [3] and Szegedy et al. [48] first suggested that neural networks may be prone to imperceptible changes in the input — the so-called adversarial perturbations — that alter the model's decision entirely. This weakness not only applies to image classification models, but is prevalent in various machine learning applications, including object detection and image segmentation [10, 54], speech recognition [8], and deep policy networks [2, 21].

The threat of adversarial perturbations has prompted tremendous effort towards the development of defense mechanisms. Common defenses either attempt to recover the true semantic labels of the input [5, 12, 19, 38, 41, 45] or detect and reject adversarial examples [17, 28, 31, 33–35, 55]. Although many of the proposed defenses have been successful against passive attackers — ones that are unaware of the presence of the defense mechanism — almost all fail against adversaries that have full knowledge of the internal details of the defense and modify the attack algorithm accordingly [1, 6]. To date, the success of existing defenses have been limited to simple datasets with relatively low variety of classes [24, 29, 39, 44, 52].

Recent studies [13, 42] have shown that the existence of adversarial perturbations may be an inherent property of natural data distributions in high dimensional spaces — painting a grim picture for defenses. However, in this paper we propose a radically new approach to defenses against adversarial attacks that turns this seemingly insurmountable obstacle from a weakness into a strength: We use the inherent property of the existence of valid adversarial perturbations around a natural image as a *signature* to attest that it is unperturbed.

---

[*]Equal Contribution. [†]Department of Computer Science, Cornell University. [‡]Department of Computer Science and Engineering, The Ohio State University. Email: {ty367, sh797, cg563, kqw4}@cornell.edu, chao.209@osu.edu.

Concretely, we exploit two seemingly contradicting properties of natural images: On one hand, natural images lie with high probability near the decision boundary to any given label [13, 42]; on the other hand, natural images are robust to random noise [48], which means these small "pockets" of spaces where the input is misclassified have low density and are unlikely to be found through random perturbations. To verify if an image is benign, we can test for both properties effectively:

1. We measure the degree of robustness to random noise by observing the change in prediction after adding i.i.d. Gaussian noise.

2. We measure the proximity to a decision boundary by observing the number of gradient steps required to change the label of an input image. This procedure is identical to running a gradient-based attack algorithm against the input (which is potentially an adversarial image already).

We hypothesize that artificially perturbed images mostly violate at least one of the two conditions. This gives rise to an effective detection mechanism even when the adversary has full knowledge of the defense. Against strong $L_\infty$-bounded white-box adversaries that adaptively optimize against the detector, we achieve a worst-case detection rate of $49\%$ at a false positive rate of $20\%$ on ImageNet [11] using a pre-trained ResNet-101 model [20]. Prior art achieves a detection rate of $0\%$ at equal false positive rate under the same setting. Further analysis shows that there exists a fundamental trade-off for white-box attackers when optimizing to satisfy the two detection criteria. Our method creates new challenges for the search of adversarial examples and points to a promising direction for future research in defense against white-box adversaries.

## 2 Background

**Attack overview.** Test-time attacks via adversarial examples can be broadly categorized into either black-box or white-box settings. In the black-box setting, the adversary can only access the model as an oracle, and may receive continuous-valued outputs or only discrete classification decisions [9, 18, 22, 23, 30, 37, 49–51]. We focus on the white-box setting in this paper, where the attacker is assumed to be an insider and therefore has full knowledge of internal details of the network. In particular, having access to the model parameters allows the attacker to perform powerful first-order optimization attacks by optimizing an adversarial loss function.

The white-box attack framework can be summarized as follows. Let $h$ be the target classification model that, given any input $\mathbf{x}$, outputs a vector of probabilities $h(\mathbf{x})$ with $h(\mathbf{x})_{y'} = p(y'|\mathbf{x})$ (i.e. the $y'$-th component of the vector $h(\mathbf{x})$) for every class $y'$. Let $y$ be the true class of $\mathbf{x}$ and $\mathcal{L}$ be a continuous-valued *adversarial loss* that encourages misclassification, e.g.,

$$\mathcal{L}(h(\mathbf{x}'), y) = -\text{cross-entropy}(h(\mathbf{x}'), y).$$

Given a target image $\mathbf{x}$ for which the model correctly classifies as $\arg\max_{y'} h(\mathbf{x})_{y'} = y$, the attacker aims to solve the following optimization problem:

$$\min_{\mathbf{x}'} \mathcal{L}(h(\mathbf{x}'), y) \text{, s.t. } \|\mathbf{x} - \mathbf{x}'\| \le \tau.$$

Here, $\|\cdot\|$ is a measure of perceptible difference and is commonly approximated using the Euclidean norm $\|\cdot\|_2$ or the max-norm $\|\cdot\|_\infty$, and $\tau > 0$ is a perceptibility threshold. This optimization problem defines an *untargeted attack*, where the adversary's goal is to cause misclassification. In contrast, for a *targeted attack*, the adversary is given some target label $y_t \ne y$ and defines the adversarial loss to encourage classification to the target label:

$$\mathcal{L}(h(\mathbf{x}'), y_t) = \text{cross-entropy}(h(\mathbf{x}'), y_t). \tag{1}$$

For the remainder of this paper, we will focus on the targeted attack setting but any approach can be readily augmented for untargeted attacks as well.

**Optimization.** White-box (targeted) attacks mainly differ in the choice of the adversarial loss functions $\mathcal{L}$ and the optimization procedures. One of the earliest attacks [48] used L-BFGS to optimize the cross-entropy adversarial loss in Equation 1. Carlini and Wagner [7] investigated the use of different adversarial loss functions and found that the margin loss

$$\mathcal{L}(Z(\mathbf{x}'), y_t) = \left[ \max_{y' \ne y_t} Z(\mathbf{x}')_{y'} - Z(\mathbf{x}')_{y_t} + \kappa \right]_+ \tag{2}$$

is more suitable for first-order optimization methods, where $Z$ is the logit vector predicted by the model and $\kappa > 0$ is a chosen margin constant. This loss is optimized using Adam [25], and the resulting method is known as the Carlini-Wagner (CW) attack. Another class of attacks favors the use of simple gradient descent using the sign of the gradient [16, 27, 32], which results in improved transferability of the constructed adversarial examples from one classification model to another.

**Enforcing perceptibility constraint.** For common choices of the measures of perceptibility, the attacker can either fold the constraint as a Lagrangian penalty into the adversarial loss, or apply a projection step at the end of every iteration onto the feasible region. Since the Euclidean norm $\|\cdot\|_2$ is differentiable, it is commonly enforced with the former option, i.e.,

$$\min_{\mathbf{x}'} \mathcal{L}(h(\mathbf{x}'), y_t) + c\|\mathbf{x} - \mathbf{x}'\|_2$$

for some choice of $c > 0$. On the other hand, the max-norm $\|\cdot\|_\infty$ is often enforced by restricting every coordinate of the difference $\mathbf{x} - \mathbf{x}'$ to the range $[-\tau, \tau]$ after every gradient step. In addition, since all pixel values must fall within the range $[0, 1]$, most methods also project $\mathbf{x}'$ to the unit cube at the end of every iteration [7, 32]. When using this option along with the cross entropy adversarial loss, the resulting algorithm is commonly referred to as the Projected Gradient Descent (PGD) attack[1] [1].

## 3   Detection Methods and Their Insufficiency

One commonly accepted explanation for the existence of adversarial examples is that they operate outside the natural image manifold — regions of the space that the model had no exposure to during training time and hence its behavior can be manipulated arbitrarily. This view casts the problem of defending against adversarial examples as a robust classification or anomaly detection problem. The former aims to project the input back to the natural image manifold and recover its true label, whereas the latter only requires determining whether the input belongs to the manifold and reject it if not.

**Detection methods.** Many principled detection algorithms have been proposed to date [17, 28, 31, 33–35, 55]. The most common approach involves testing the input against one or several criteria that are satisfied by natural images but are likely to fail for adversarially perturbed images. In what follows, we briefly describe two representative detection mechanisms.

*Feature Squeezing* [55] applies a semantic-preserving image transformation to the input and measures the difference in the model's prediction compared to the plain input. Transformations such as median smoothing, bit quantization, and non-local mean do not alter the image content; hence the model is expected to output similar predictions after applying these transformations. The method then measures the maximum $L_1$ change in predicted probability after applying these transformations and flags the input as adversarial if this change is above a chosen threshold.

*Artifacts* [14] uses the empirical density of the input and the model uncertainty to characterize benign and adversarial images. The empirical density can be computed via kernel density estimation on the feature vector. For the uncertainty estimate, the method evaluates the network multiple times using different random dropout masks and computes the variance in the output. Under the Bayesian interpretation of dropout, this variance estimate encodes the model's uncertainty [15]. Adversarial inputs are expected to have lower density and higher uncertainty than natural inputs. Thus, the method predicts the input as adversarial if these criteria are below or above a chosen threshold.

Detectors that use multiple criteria (such as Feature Squeezing and Artifacts) can combine these criteria into a single detection method by either declaring the input as adversarial if any criterion fails to be satisfied, or by training a classifier on top of them as features to classify the input. Other notable useful features for detecting adversarial images include convolutional features extracted from intermediate layers [28, 34], distance to training samples in pixel space [17, 31], and entropy of non-maximal class probabilities [36].

**Bypassing detection methods.** While the approaches for detecting adversarial examples appear principled in nature, the difference in settings from traditional anomaly detection renders most techniques easy to bypass. In essence, a white-box adversary with knowledge of the features used for detection can optimize the adversarial input to mimic these features with gradient descent. Any non-differentiable component used in the detection algorithm, such as bit quantization and non-local mean,

can be approximated with the identity transformation on the backward pass [1], and randomization can be circumvented by minimizing the expected adversarial loss via Monte Carlo sampling [1]. These simple techniques have proven tremendously successful, bypassing almost all known detection methods to date [6]. Given enough gradient queries, adversarial examples can be optimized to appear even "more benign" than natural images.

# 4 Detection by Adversarial Perturbations

In this section we describe a novel approach to detect adversarial images that relies on two principled criteria regarding *the distribution of adversarial perturbations around natural images*. In contrast to the shortcomings of prior work, our approach is hard to fool through first-order optimization.

## 4.1 Criterion 1: Low density of adversarial perturbations

The features extracted by convolutional neural networks (CNNs) from natural images are known to be particularly robust to random input corruptions [19, 48, 53]. In other words, random perturbations applied to natural images should not lead to changes in the predicted label (i.e. an adversarial image). Our first criterion follows this intuition and tests if the given input is robust to Gaussian noise:

**C1: Robustness to random noise.** Sample $\epsilon \sim N(0, \sigma^2 I)$ (where $\sigma^2$ is a hyperparameter) and compute $\Delta = \|h(\mathbf{x}) - h(\mathbf{x} + \epsilon)\|_1$. The input $\mathbf{x}$ is rejected as adversarial if $\Delta$ is sufficiently large.

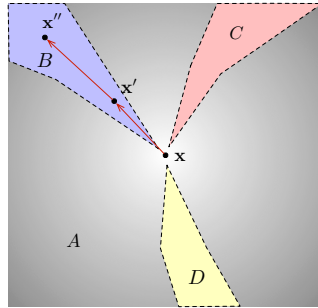

This style of reasoning has indeed been successfully applied to defend against black-box and gray-box[2] attacks [19, 40, 53]. Figure 1 shows a 2D cartoon depiction of the high dimensional decision boundary near a natural image $\mathbf{x}$. When the adversarial attack perturbs $\mathbf{x}$ slightly across the decision boundary from A to an incorrect class B, the resulting adversarial image $\mathbf{x}'$ can be easily randomly perturbed to return to class A and will therefore fail criterion C1.

However, we emphasize that this criterion alone is insufficient against white-box adversaries and can be easily bypassed. In order to make the adversarial image also robust against Gaussian noise, the attacker can optimize the expected adversarial loss under this defense strategy [1] through Monte Carlo sampling of noise vectors during optimization. This effectively produces an adversarial image $\mathbf{x}''$ (see Figure 1) that is deep inside the decision boundary.

Figure 1: Schematic illustration of the shape of adversarial regions near a natural image $\mathbf{x}$.

More precisely, for a natural image $\mathbf{x}$ with correctly predicted label $y$ and target label $y_t$, let $h(\mathbf{x})$ be the predicted class-probability vector. Let us define $\mathbf{p}^{\text{adv}}$ to be identical to $h(\mathbf{x})$ in every dimension, except for the correct class $y$ and the target $y_t$, where the two probabilities are swapped. Consequently, dimension $y_t$ is the dominant prediction in $\mathbf{p}^{\text{adv}}$. We redefine the adversarial loss of the (targeted) PGD attack to contain two terms:

$$\mathcal{L}^\star = \mathcal{L}_1 + \mathcal{L}_2 \text{ where: } \mathcal{L}_1 = \underbrace{\mathcal{L}(h(\mathbf{x}'), \mathbf{p}^{\text{adv}})}_{\text{misclassify } \mathbf{x}' \text{ as } y_t}, \text{ and } \mathcal{L}_2 = \underbrace{\mathbb{E}_{\epsilon \sim N(0, \sigma^2 I)} \left[ \|h(\mathbf{x}') - h(\mathbf{x}' + \epsilon)\|_1 \right]}_{\text{bypass C1}}, \quad (3)$$

where $\mathcal{L}(\cdot, \cdot)$ denotes the cross-entropy loss. For the first term, we deviate from standard attacks by targeting the probability vector $\mathbf{p}^{\text{adv}}$ instead of the one-hot vector corresponding to label $y_t$. Optimizing against the one-hot vector would cause the adversarial example to over-saturate in probability, which artificially increases the difference $\Delta = \|h(\mathbf{x}') - h(\mathbf{x}' + \epsilon)\|_1$ and makes it easier to detect using criterion C1.

We evaluate this white-box attack against criterion C1 using a pre-trained ResNet-101 [20] model on ImageNet [11] as the classification model. We sample 1,000 images from the ImageNet validation set and optimize the adversarial loss $\mathcal{L}^\star$ for each of them using Adam [25] with learning rate 0.005 for a maximum of 400 steps to construct the adversarial images.

Figure 2 (left) shows the effect of the number of gradient iterations on $\Delta$ when optimizing the adversarial loss $\mathcal{L}^\star$. The center line shows median values of $\Delta$ across 1,000 sample images, and

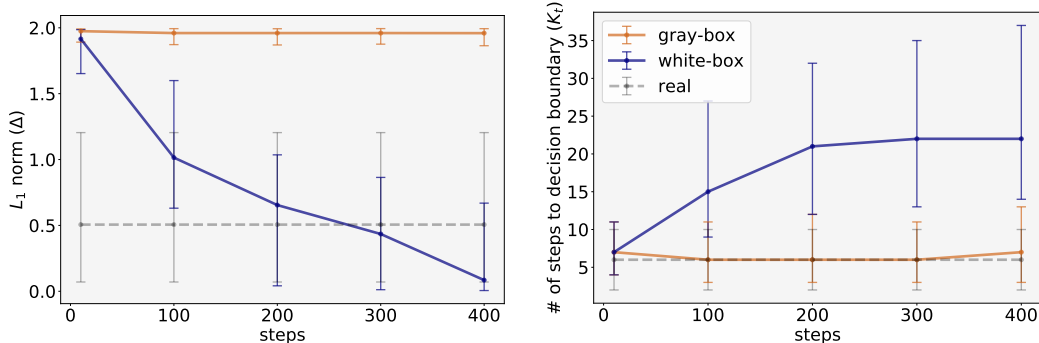

Figure 2: The variation in $\Delta$ under Gaussian perturbations (C1; left plot) and numbers of steps $K_t$ to the decision boundary (C2t; right plot) for adversarial images constructed using different numbers of gradient iterations. Gray-box attacks (orange) can be detected easily with criterion C1 alone (left plot, the orange line is significantly higher than the gray line). For white-box attacks (blue), C1 alone is not sufficient (the blue line overlaps with the gray line) — however C2 (right plot) separates the two lines reliably when C1 does not.

the error bars show the range of values between the 30th and 70th quantiles. When the attacker is agnostic to the detector (orange line), i.e., only optimizing $\mathcal{L}_1$, $\Delta$ does not decrease throughout optimization and can be used to perfectly separate adversarial and real images (gray line). However, in the white-box attack, the adversarial loss explicitly encourages $\Delta$ to be small, and we observe that indeed the blue line shows a downward trend as the adversary proceeds through gradient iterations. As a result, the range of values for $\Delta$ quickly begins to overlap with and fall below that of real images after 100 steps, which shows that criterion C1 alone cannot be used to detect adversarial examples.

## 4.2 Criterion 2: Close proximity to decision boundary

The intuitive reason why the attack strategy described above in section 4.1 can successfully fool criterion C1 is that it effectively pushes the adversarial image far into the decision boundary of the target class (e.g. $\mathbf{x}''$ in Figure 1) — an unlikely position for a natural image, which tends to be close to adversarial decision boundaries. Indeed, Fawzi et al. [13] and Shafahi et al. [42] have shown that adversarial examples are inevitable in high-dimensional spaces. Their theoretical arguments suggest that, due to the curse of dimensionality, a sample from the natural image distribution is close to the decision boundary of any classifier with high probability. Hence, we define a second criterion to test if an image is close to the decision boundary of an incorrect class:

**C2(t/u): Susceptibility to adversarial noise.** For a chosen first-order iterative attack algorithm $\mathcal{A}$, evaluate $\mathcal{A}$ on the input $\mathbf{x}$ and record the minimum number of steps $K$ required to adversarially perturb $\mathbf{x}$. The input is rejected as adversarial if $K$ is sufficiently large.

Criterion C2 can be further specialized to *targeted* attacks (C2t) and *untargeted* attacks (C2u), which measures the proximity (i.e. number of gradient steps) to either a chosen target class or to an arbitrary but different class. We denote these quantities as $K_t$ and $K_u$, respectively. In this paper we choose $\mathcal{A}$ in C2 to be the targeted/untargeted PGD attack, but our framework can plausibly generalize to any first-order attack algorithm. Figure 2 (right) shows the effect of optimizing the adversarial loss $\mathcal{L}^\star$ on $K_t$. Again, the center line shows the median value of $K_t$ across 1,000 images and the error bars indicate the 30th and 70th quantiles. As expected, real images (gray line) require very few steps to reach the decision boundary of any random target class. When the adversary does not seek to bypass criterion C1 (orange line), the constructed adversarial images lie very close to the decision boundary and are indistinguishable from real images with C2 alone (however here C1 is already sufficient).

On the other hand, when the attacker minimizes $\Delta$ to fool criterion C1, the adversarial image moves away from the decision boundary in order to be robust to random Gaussian noise. This results in an increase in the number of steps $K_t$ to reach the decision boundary of a random target class. At 400 steps, there is almost no overlap between the 30-70th quantiles of values of $K_t$ for real and adversarial images. This separation begins almost precisely as the value of $\Delta$ for adversarial images (left plot) begins to overlap with that of natural images at 100 steps. Thus, C2t becomes an effective criterion to detect adversarial images that optimize against C1.

### 4.3 Detection strategy

The fact that natural images can simultaneously satisfy criteria C1 and C2 can be regarded as almost paradoxical: While the minimum distance from a natural input to the decision boundary of any incorrect class is small, the density of directions that can lead to a decision boundary within a short distance is also very low. We postulate that this behavior of natural images is difficult to imitate even for an adaptive, white-box adversary.

Our detection strategy using the two criteria can be summarized as follows. Given an input $\mathbf{x}$ (which might be an adversarial example already), we compute $(\Delta, K_t, K_u)$ and compare these quantities to chosen thresholds $(t_{C1}, t_{C2t}, t_{C2u})$, corresponding to criteria C1, C2t, and C2u. We reject $\mathbf{x}$ as an adversarial example if at least one of the three (sub-)criteria is not satisfied, i.e., if any measurement is larger than the corresponding threshold. Details on hyperparameter selection can be found in the Supplementary Material.

**Best effort white-box adversary.** Based on our proposed detection method, we define a white-box adversary that aims to cause misclassification while passing the detection criteria C1 and C2. Let $\mathcal{L}$ be the adversarial loss for the defense-agnostic (targeted) attack (e.g. Equation 1). We define loss functions $\mathcal{L}_1$ and $\mathcal{L}_2$ as in Equation 3 following the same strategy used in section 4.1 to bypass C1. Since the criterion C2t is discrete, it is difficult to optimize directly. Instead, we encourage the constructed adversarial image to change prediction to *any* class $y' \neq y_t$ after a *single* gradient step towards $y'$. As natural images require very few gradient steps to cross the decision boundary, the resulting adversarial image will appear real to criterion C2t. Let

$$\delta_{y'} = \nabla_{\mathbf{x}'} \mathcal{L}_{\mathcal{A}}(h(\mathbf{x}'), y')$$

denote the gradient of the cross-entropy loss w.r.t. $\mathbf{x}'$[3]. The loss term to bypass C2t can be defined as

$$\mathcal{L}_3 = \mathbb{E}_{y' \sim \text{Uniform}, y' \neq y_t}[\mathcal{L}(h(\mathbf{x}' - \alpha\delta_{y'}), y')],$$

which encourages $\mathbf{x}' - \alpha\delta_{y'}$ — the one-step move towards class $y'$ at step size $\alpha$ — to be close to or cross the decision boundary of class $y'$ for every randomly chosen class $y' \neq y_t$. Similarly, to bypass criterion C2u, we simulate one gradient step at step size $\alpha$ *away from* the target class $y_t$ (which the defender perceives as the predicted class) as $\mathbf{x}' + \alpha\delta_{y_t}$. We then encourage this resulting image to be classified as not $y_t$ via the loss term:

$$\mathcal{L}_4 = -\mathcal{L}(h(\mathbf{x}' + \alpha\delta_{y_t}), y_t).$$

Gradients for $\mathcal{L}_3$ and $\mathcal{L}_4$ can be approximated using Backward Pass Differentiable Approximation (BPDA) [1]. As a result of optimizing $\mathcal{L}_3$ and $\mathcal{L}_4$, the produced image $\mathbf{x}'$ will admit both a targeted and an untargeted "adversarial example" within one or few steps of the attack algorithm $\mathcal{A}$, therefore bypassing C2. Combining all the components, the modified adversarial loss $\mathcal{L}^\star$ for white-box attack against our detector becomes

$$\mathcal{L}^\star = \lambda\mathcal{L}_1 + \mathcal{L}_2 + \mathcal{L}_3 + \mathcal{L}_4. \tag{4}$$

The inclusion of additional loss terms hinders the optimality of $\mathcal{L}_1$ and may cause the attack to fail to generate a valid adversarial example. Thus, we include the coefficient $\lambda$ so that $\mathcal{L}_1$ dominates the other loss terms and guarantees close to $100\%$ success rate in constructing adversarial examples to fool $h$. We optimize the total loss $\mathcal{L}^\star$ using Adam [25].

## 5 Experiments

We test our detection mechanism against the white-box attack defined in section 4.3 in several different settings, and release our code publicly for reproducibility[4].

### 5.1 Setup

**Datasets and target models.** We conduct our empirical studies on ImageNet [11] and CIFAR-10 [26]. We sample 1,000 images from ImageNet (validation) and CIFAR-10 (test): each class has 1 or 100 images. We use the pre-trained ResNet-101 model [20] in PyTorch for ImageNet and train a VGG-19 model [43] with a dropout rate of 0.5 [46] for CIFAR-10 as target models. We additionally include detection results using an Inception-v3 model [47] on ImageNet in the Supplementary Material.

Table 1: Detection rates of different detection algorithms against white-box adversaries on ImageNet. Worst-case performance against all evaluated attacks is underlined for each detector.

| Detector | FPR | PGD | | | CW | | |
|---|---|---|---|---|---|---|---|
| | | LR=0.01 | LR=0.03 | LR=0.1 | LR=0.01 | LR=0.03 | LR=0.1 |
| Feature Squeezing | 0.2 | | 0.003 | | | 0.000 | |
| Feature Squeezing | 0.1 | | 0.002 | | | 0.000 | |
| C1 | 0.2 | 0.585 | 0.132 | 0.066 | 0.682 | 0.103 | 0.068 |
| C2t | 0.2 | 0.205 | 0.649 | 0.724 | 0.436 | 0.800 | 0.882 |
| C2u | 0.2 | 0.001 | 0.001 | 0.002 | 0.154 | 0.042 | 0.039 |
| Combined | 0.2 | 0.494 | 0.490 | 0.612 | 0.688 | 0.718 | 0.809 |
| C1 | 0.1 | 0.320 | 0.043 | 0.013 | 0.486 | 0.044 | 0.021 |
| C2t | 0.1 | 0.120 | 0.483 | 0.616 | 0.287 | 0.709 | 0.806 |
| C2u | 0.1 | 0.000 | 0.000 | 0.000 | 0.062 | 0.010 | 0.003 |
| Combined | 0.1 | 0.269 | 0.264 | 0.378 | 0.512 | 0.482 | 0.601 |

**Attack algorithms.** We evaluate our detection method against the white-box adversary defined in section 4.3. Since the adversary may vary in the choice of the surrogate loss (cf. $\mathcal{L}$ in Equation 3), we experiment using both targeted and untargeted variants of two representative loss functions: the margin loss defined in the Carlini-Wagner (CW) attack [7] (see Equation 2), and the cross-entropy loss used in the Projected Gradient Descent (PGD) attack [1]. The $L_\infty$-bound for all attacks is set to $\tau = 0.1$, which is very strong and often produces images with noticeable visual distortion. See Figure 3 for an illustration. We further experiment with boundary attack [4] for attacking the target model and detection mechanism as a black box in the Supplementary Material.

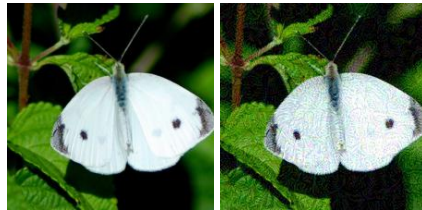

Figure 3: A sample clean (left) and adversarial (right) image at $L_\infty$ perceptibility threshold of $\tau = 0.1$.

All attacks optimize the adversarial loss using Adam [25]. We set $\lambda = 2$ (cf. Equation 4) for ImageNet and $\lambda = 3$ for CIFAR-10 to guarantee close to $100\%$ attack success rate. We found that changing the maximum number of iterations has little effect on the attack's ability to bypass our detector, and thus we fix to a reasonable value of 50 steps for ImageNet (which is sufficient to guarantee convergence; see Figure 4) and 200 steps for CIFAR-10. The learning rate has a more noticeable effect and we evaluate our detector against different chosen values. See the Supplementary Material for detection results against variants of these attacks, including untargeted attacks and $\tau = 0.03$.

**Baselines.** We compare our detector against two strategies: Feature Squeezing [55] and Artifacts [14]. These detection algorithms are the most similar to ours — using a combination of different criteria as features for the detector. We modify the Artifacts defense slightly to use the density and uncertainty estimates directly by thresholding rather than training a classifier on top of these features, which has been shown in prior work [6] to remain effective against adversaries that are agnostic to the defense. With a false positive rates (FPR) of 0.1, Feature Squeezing attains a detection rate of 0.737 on ImageNet and 0.892 on CIFAR-10, while Artifacts attains a detection rate of 0.587 on CIFAR-10.

We adopt the same strategy as in section 4.3 to formulate white-box attacks against these detectors, adding a term in the adversarial loss for each criterion and using Backward Pass Differentiable Approximation (BPDA) to compute the gradient of non-differentiable transformations [1]. Details on these modifications can be found in the Supplementary Material.

## 5.2 Detection results

**ImageNet results.** Table 1 shows the detection rate of our method against various adversaries on ImageNet. We evaluate our detector under two different settings, resulting in FPR of 0.1 and 0.2. Entries in the table correspond to the detection rate (or true positive rate) when the white-box adversary defined in section 4.3 is applied to attack the model along with the detector.

Under all six attack settings (PGD vs. CW, LR = $0.01, 0.03, 0.1$), our detector performs substantially better than random, achieving a worst-case detection rate of 0.49 at FPR = 0.2 and 0.264 at FPR =

Table 2: Detection rates of different detection algorithms against white-box adversaries on CIFAR-10. Worst-case performance against all evaluated attacks is underlined for each detector.

| Detector | FPR | PGD | | | CW | | |
|---|---|---|---|---|---|---|---|
| Feature Squeezing | 0.2 | 0.074 | | | 0.096 | | |
| Feature Squeezing | 0.1 | 0.008 | | | 0.021 | | |
| Artifacts | 0.2 | 0.108 | | | 0.018 | | |
| Artifacts | 0.1 | 0.090 | | | 0.009 | | |
| | | LR=0.001 | LR=0.01 | LR=0.1 | LR=0.001 | LR=0.01 | LR=0.1 |
| C1 | 0.2 | 1.000 | 0.991 | 0.792 | 0.422 | 0.033 | 0.012 |
| C2t | 0.2 | 0.024 | 0.050 | 0.346 | 0.098 | 0.786 | 0.971 |
| C2u | 0.2 | 0.000 | 0.000 | 0.000 | 0.000 | 0.000 | 0.000 |
| Combined | 0.2 | 0.998 | 0.984 | 0.660 | 0.374 | 0.481 | 0.740 |
| C1 | 0.1 | 0.986 | 0.953 | 0.207 | 0.283 | 0.016 | 0.007 |
| C2t | 0.1 | 0.010 | 0.015 | 0.180 | 0.026 | 0.581 | 0.858 |
| C2u | 0.1 | 0.000 | 0.000 | 0.000 | 0.000 | 0.000 | 0.000 |
| Combined | 0.1 | 0.966 | 0.909 | 0.187 | 0.263 | 0.356 | 0.568 |

0.1 on ImageNet. This result is a considerable improvement over similar detection methods such as Feature Squeezing, where the detection rate is close to 0, i.e. the adversarial images appear "more real" than natural images. We emphasize that given the strong adversary that we evaluate against ($\tau = 0.1$), these detection rates are very difficult to attain against white-box attacks.

**Ablation study.** We further decompose the components of our detector to demonstrate the trade-offs the adversary must make when attacking our detector. When using different learning rates, the adversary switches between attempting to fool criteria C1 and C2. For example, at LR = 0.01, the PGD adversary can be detected using criterion C1 substantially better than using criterion C2t due to under-optimization of the value $\Delta$. On the other hand, at LR = 0.1, the adversary succeeds in bypassing criterion C1 at the cost of failing C2t. The criterion C2u does not appear to be effective here as it consistently achieves a detection rate of close to 0. However, it is a crucial component of our method against untargeted attacks (see Supplementary Material). Overall, our combined detector achieves the best worst-case detection rate across all attack scenarios.

**CIFAR-10 results.** The detection rates for our method are slightly worse on CIFAR-10 (Table 2) but still outperforming the Feature Squeezing and Artifacts baselines, which are close to 0 in the worst case. For this dataset, criterion C2u becomes ineffective due to the over-saturation of predicted probabilities for clean images, causing untargeted perturbation to take excessively many steps.

Furthermore, the CIFAR-10 dataset violates both of our hypotheses regarding the distribution of adversarial perturbations near a natural image. Models trained on CIFAR-10 are much less robust to random Gaussian noise due to lack of data augmentation and poor diversity of training samples — the VGG-19 model could only tolerate a Gaussian noise of $\sigma = 0.01$ as opposed to $\sigma = 0.1$ for ResNet-101 on ImageNet. Furthermore, CIFAR-10 is much lower-dimensional than ImageNet, hence natural images are comparatively farther from the decision boundary [13, 42]. Given this observation, we suggest that our detector be used only in situations where these two assumptions can be satisfied.

**Gray-box detection results.** Despite the fact that our detection mechanism is formulated against white-box adversaries, we evaluated against a gray-box adversary with knowledge of the underlying model but not of the detector for completeness.

Table 3 shows detection rates for gray-box attacks at FPR of 0.05 and 0.1 on ImageNet. At perceptibility bound $\tau = 0.03$, the combined detector is very successful at detecting the gener-

Table 3: Detection rates for variations of the gray-box adversary on ImageNet. Worst-case performance against all evaluated attacks is underlined for each detector.

| Detector | FPR | $\tau = 0.03$ | | $\tau = 0.1$ | |
|---|---|---|---|---|---|
| | | PGD | CW | PGD | CW |
| Feature Squeezing | 0.05 | 0.669 | 0.304 | 0.572 | 0.014 |
| Feature Squeezing | 0.1 | 0.758 | 0.336 | 0.672 | 0.020 |
| **Ours**: Combined | 0.05 | 0.976 | 0.981 | 0.896 | 0.570 |
| **Ours**: Combined | 0.1 | 0.990 | 0.989 | 0.915 | 0.678 |

ated adversarial images, achieving a detection rate of $97.6\%$ at $5\%$ FPR. In comparison, Feature Squeezing could only achieve a detection rate of $30.4\%$ against the CW attack. Against the much

stronger adversary at $\tau = 0.1$, both detectors perform significantly worse, but our combined detector still achieves a non-trivial detection rate.

## 5.3 Adversarial loss curves

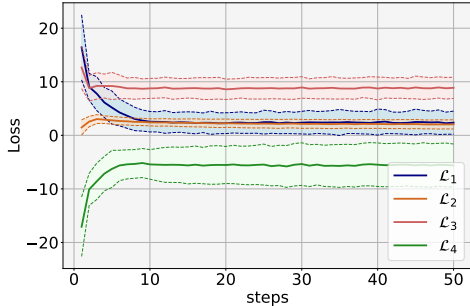

Figure 4: Plot of different components of the adversarial loss $\mathcal{L}^*$. See text for details.

To further substantiate our claim that the criteria C1 and C2t/u are mutually exclusive, we plot the value of different components of the adversarial loss $\mathcal{L}^*$ throughout optimization for the white-box attack (PGD) on ImageNet. The center lines in Figure 4 show the average loss for each $\mathcal{L}_i$ over 1,000 images and the shaded areas indicate standard deviation. Since the primary goal is to cause misclassification, the term $\mathcal{L}_1$ (blue line) shows steady descending trend throughout optimization and its value has stabilized after 50 iterations. $\mathcal{L}_2$ (orange line) begins at a low value due to the initialization being a natural image (and hence it is robust against Gaussian noise), and after 50 iterations it returns back to the initial level, which shows that the adversary is successful at bypassing criterion C1. However, this success comes at the cost of $\mathcal{L}_3$ (red line) failing to reduce to a sufficiently low level due to inherent conflict with $\mathcal{L}_2$ (and $\mathcal{L}_1$), hence criterion C2t can be used to detect the resulting adversarial image.

## 5.4 Detection times

One drawback of our method is its (relatively) high computation cost. Criteria C2t/u require executing a gradient-based attack until either label change or for a specified number of steps. To limit the number of false positives, the upper threshold on the number of gradient steps must be sufficiently high, dominating the running time of the detection algorithm. Table 4 shows the average per-image detection time for both real and (targeted) adversarial images on ImageNet and CIFAR-10. On both datasets, the average detection time for real images is approx-

Table 4: Running time of different components of our detection algorithm on ImageNet and CIFAR-10. See text for details.

|          |      | Real    | PGD    | CW     |
|----------|------|---------|--------|--------|
|          | C1   | 0.074s  | 0.091s | 0.107s |
| ImageNet | C2t  | 0.403s  | 1.057s | 3.46s  |
|          | C2u  | 4.512s  | 0.138s | 0.241s |
|          | C1   | 0.011s  | 0.013s | 0.012s |
| CIFAR-10 | C2t  | 0.379s  | 0.128s | 0.27s  |
|          | C2u  | 5.230s  | 0.055s | 9.631s |

imately 5 seconds and is largely due to a large threshold for C2u. The situation is similar for adversarial images: As the CW attack optimizes the margin loss, taking the adversarial images much farther into the decision boundary, it takes longer (many more steps to undo via C2t/u) to detect it.

## 6 Conclusion

We have shown that our detection method achieves substantially improved resistance to white-box adversaries compared to prior work. In contrast to other detection algorithms that combine multiple criteria, the criteria used in our method are mutually exclusive — optimizing one will negatively affect the other — yet are inherently true for natural images. While we do not suggest that our method is impervious to white-box attacks, it does present a significant hurdle to overcome and raises the bar for any potential adversary.

There are, however, some limitations to our method. The running time of our detector is dominated by testing criterion C2, which involves running an iterative gradient-based attack algorithm. The high computation cost could prohibit the suitability of our detector for deployment. Furthermore, it is fair to say that the false positive rate remains relatively high due to a large variance in the statistics $\Delta$, $K_t$ and $K_u$ for the different criteria, hence a threshold-based test cannot completely separate real and adversarial inputs. Future research that improve in either front can certainly ameliorate the performance of our method to be more practical in real world systems.

**Acknowledgments**

C.G., W-L.C., K.Q.W. are supported by grants from the NSF (III-1618134, III-1526012, IIS-1149882, IIS-1724282, and TRIPODS-1740822), the Bill and Melinda Gates Foundation, and the Cornell Center for Materials Research with funding from the NSF MRSEC program (DMR-1719875); and are also supported by Zillow, SAP America Inc., and Facebook. We thank Pin-Yu Chen (IBM) for constructive discussions.

## Footnotes

[1]Some literature also refer to the iterative Fast Gradient Signed Method (FGSM) [16] as PGD [32].

[2]In gray-box attacks, the adversary has full access to the classifier $h$ but is agnostic to the defense mechanism.

[3]We denote the adversarial loss of the Algo. $\mathcal{A}$ in our detector by $\mathcal{L}_{\mathcal{A}}$ to differentiate it from $\mathcal{L}$ of the attacker.

[4]https://github.com/s-huu/TurningWeaknessIntoStrength

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
