[Supplementary Material]

# Supplementary Material: A New Defense Against Adversarial Images: Turning a Weakness into a Strength

## A  Additional Experiments

### A.1  Detection rates on Inception network

Table 5 shows detection rates on ImageNet using a Inception-v3 model [48] by criteria C1, C2t, and C2u individually and jointly. We observe an almost identical trend as using ResNet-101 as the target model (Table 1 in main paper): the adversary cannot simultaneously fool both criteria C1 and C2t. Detection rates by Feature Squeezing are slightly higher than those for ResNet-101 but remain close to 0 and are substantially worse than those by our combined detector.

Table 5: Detection rates for different detection algorithms against white-box adversaries on ImageNet with Inception-v3 target model. Worst-case performance against all evaluated attacks is underlined for each detector.

| Detector | FPR | PGD | | | CW | | |
|---|---|---|---|---|---|---|---|
| Feature Squeezing | 0.2 | | 0.068 | | | 0.031 | |
| Feature Squeezing | 0.1 | | 0.062 | | | 0.013 | |
| | | LR=0.01 | LR=0.03 | LR=0.1 | LR=0.01 | LR=0.03 | LR=0.1 |
| C1 | 0.2 | 0.858 | 0.712 | 0.628 | 0.803 | 0.483 | 0.362 |
| C2t | 0.2 | 0.173 | 0.411 | 0.424 | 0.449 | 0.585 | 0.543 |
| C2u | 0.2 | 0.004 | 0.013 | 0.003 | 0.346 | 0.225 | 0.067 |
| Combined | 0.2 | 0.762 | 0.546 | 0.468 | 0.788 | 0.527 | 0.479 |
| C1 | 0.1 | 0.648 | 0.36 | 0.258 | 0.688 | 0.29 | 0.142 |
| C2t | 0.1 | 0.043 | 0.157 | 0.18 | 0.231 | 0.322 | 0.321 |
| C2u | 0.1 | 0.001 | 0.006 | 0.003 | 0.255 | 0.114 | 0.056 |
| Combined | 0.1 | 0.516 | 0.257 | 0.203 | 0.635 | 0.281 | 0.199 |

Table 6: Detection rates for variations of the white-box adversary. Worst-case performance against all evaluated attacks is underlined for each detector.

| Detector | FPR | PGD | | | CW | | |
|---|---|---|---|---|---|---|---|
| | | LR=0.01 | LR=0.03 | LR=0.1 | LR=0.01 | LR=0.03 | LR=0.1 |
| Small radius ($\tau = 0.03$) | 0.2 | 0.715 | 0.674 | 0.571 | 0.934 | 0.86 | 0.713 |
| Small radius ($\tau = 0.03$) | 0.1 | 0.583 | 0.522 | 0.418 | 0.894 | 0.753 | 0.500 |
| $\mathcal{L}^* = \mathcal{L}_1 + \mathcal{L}_2$ | 0.2 | 0.695 | 0.765 | 0.800 | 0.738 | 0.604 | 0.512 |
| $\mathcal{L}^* = \mathcal{L}_1 + \mathcal{L}_2$ | 0.1 | 0.527 | 0.572 | 0.632 | 0.58 | 0.353 | 0.304 |
| Untargeted Attack | 0.2 | 0.994 | 0.997 | 0.997 | 0.538 | 0.567 | 0.576 |
| Untargeted Attack | 0.1 | 0.987 | 0.995 | 0.995 | 0.395 | 0.342 | 0.378 |

### A.2  Variations of the white-box attack

We further analyze our detection method in three different attack scenarios: Using a smaller perceptibility threshold $\tau = 0.03$, attacking criterion C1 only, and performing untargeted attack. The second variation is of interest since the losses $\mathcal{L}_3$ and $\mathcal{L}_4$ are in direct conflict with $\mathcal{L}_2$ (and $\mathcal{L}_1$), possibly hindering optimization.

Table 6 shows detection rates for the combined detector using criteria C1 and C2 against these attack variations on ResNet-101. First, as expected, we see that the small radius attack ($\tau = 0.03$ at the top two rows) is substantially easier to detect than the one with $\tau = 0.1$. When evaluated against the attack that only targets C1 (middle two rows) and against untargeted attack (last two rows), our

method remains effective and the worst-case detection rate is higher than that for the targeted attack in section 5.2. These experimental observations suggest that the white-box adversary we evaluate against in section 5.2 could be the optimal first-order attack algorithm against our detector and confirms that our evaluation protocol is sound.

## A.3 Backpropagation through the gradients

In the section 4.3, we obtain the gradients of $\mathcal{L}_3$ and $\mathcal{L}_4$ for white-box attack using Backward Pass Differentiable Approximation (BPDA) [1]. Here, we investigate backpropagation through the gradients to obtain the gradients, an approach commonly used in second order methods [15]. The results on ImageNet are reported in Table 7. The detection rates are similar to those of using BPDA (Table 1, combined) in section 5.2 except that CW attack at LR = 0.01 becomes stronger. The attack time is significantly longer: on average 31 sec/image compared to 14 sec/image by BPDA.

Table 7: Detection rates of our combined method against white-box adversaries using backpropagation through the gradients on ImageNet.

| LR | FPR | PGD | CW |
|---|---|---|---|
| 0.01 | 0.2 | 0.536 | 0.587 |
| 0.03 | 0.2 | 0.652 | 0.794 |
| 0.10 | 0.2 | 0.710 | 0.817 |
| 0.01 | 0.1 | 0.288 | 0.337 |
| 0.03 | 0.1 | 0.411 | 0.593 |
| 0.10 | 0.1 | 0.493 | 0.692 |

## A.4 Comparison to adversarial training

We compare to adversarial training [33], which aims to directly learn a target model that is robust to adversaries. We note that [33] reports recognition accuracy rather than detection rates. Besides, our results (on CIFAR) are based on an $L_\infty$ perturbation norm bound of 0.1, which is much larger than the bound of 0.03 used in [33] and makes detection much harder. To better compare against their work, we examine with the adversarially trained model that has a $87.3\%$ accuracy on natural images while the undefended model has $95.3\%$, which is close to the $10\%$ FPR setting we use in our experiments. We also evaluate our detector against the $L_\infty$-bound of 0.03. Under the strongest PGD attack, our approach has a detection rate of $84.0\%$ (50-step PGD, LR = 0.1) while the adversarially trained model has a recognition accuracy of $45.8\%$ (20-step PGD) on adversarial images.

## A.5 Boundary attack

To evaluate our detector against gradient-free adversaries, we experiment with boundary attack [4] for attacking the target model and detection mechanism as a black box. We restrict the mean squared error (MSE) of the adversarial perturbation to be less than 0.01, which is comparable to an $L_\infty$ bound of 0.1. We evaluate the attack in three scenarios: attacking criterion C1 alone, criteria C1+C2t, and criteria C1+C2t+C2u. Here a successful adversary is an image which not only fools the detector but also has MSE less than 0.01; otherwise, we consider this adversary to be detected. We plot the detection rate curve on ImageNet in Figure 5. We

Figure 5: Detection rates against boundary attack at different steps of attacks on ImageNet.

observe a similar trend against gradient-based adversaries ( Table 1 of the main text): the criterion C1 alone is insufficient for detection, while adding the criterion C2 greatly increase the difficulty of attack. When the detection criteria is C1+C2t+C2u, the mutual exclusivity of these criteria prohibits optimization progress of the boundary attack.

# B Implementation Details

## B.1 Hyperparameter settings for detector

Our detection algorithm requires the following hyperparameters:

**Criterion 1.** We set the variance parameter $\sigma$ such that predictions on real images are minimally affected after random perturbation. This quantity is set to $\sigma = 0.1$ on ImageNet and $\sigma = 0.01$ on CIFAR-10.

**Criterion 2.** Hyperparameters for criterion C2t (number of steps to a chosen target class) consist of all hyperparameters in the attack algorithm $\mathcal{A}$, including step size, maximum number of steps, and perceptibility threshold $\tau_{\mathcal{A}}$. On ImageNet, we chose a step size of 0.005, allow a maximum of 200 steps, and set $\tau_{\mathcal{A}} = 0.03$. This setting guarantees that most real images will be successfully perturbed to cross the decision boundary. The hyperparameters for C2u are different due to over-saturation of predicted probabilities. Thus, we chose a step size of 0.2 and allow a maximum of 1,000 steps. The perceptibility threshold remains at $\tau_{\mathcal{A}} = 0.03$. Hyperparameters for Inception-v3 and for VGG-19 on CIFAR-10 are set similarly but are adapted to the particular model and dataset.

## B.2 Details for white-box attack against baselines

In this section we give details for the white-box attack used against Feature Squeezing and Artifacts.

**Feature Squeezing** applies three different transformations — median smoothing, bit quantization, and non-local mean — to the input and measures the $L_1$ distance in predicted probability before and after transformation. We modify the white-box attack to bypass this defense as follows. Let $F_1, F_2, F_3$ denote the three transformations. The modified (PGD) adversarial loss is defined as

$$\mathcal{L}^{\star}_{\text{FS}} = \text{cross-entropy}(h(\mathbf{x}'), y_t) + \sum_{i=1}^{3} \left[ \|h(\mathbf{x}') - h(F_i(\mathbf{x}'))\|_1 \right].$$

Gradients of non-differentiable transformations, namely bit quantization and non-local mean, are approximated using BPDA [1].

**Artifacts** uses empirical measures of density in feature space and model uncertainty estimated using dropout to characterize adversarial examples. To bypass this defense, we can alter the adversarial loss to maximize density and minimize uncertainty while causing misclassification.

Density is computed via kernel density estimation in feature space with a Gaussian kernel. This quantity, say $\phi(\mathbf{x})$, is differentiable and can be directly optimized via gradient descent. On the other hand, minimizing uncertainty can be achieved by computing the empirical variance via Monte Carlo sampling. More specifically, let $h_{\mathbf{b}}$ be the classification model with dropout mask $\mathbf{b} \sim$ Bernoulli$(0.5)^m$, where $m$ is the number of model parameters. Each iteration, we sample $N = 50$ dropout masks $b_i$ and compute $\mathbf{p}_i = h_{\mathbf{b}_i}(\mathbf{x})$ for $i = 1, \ldots, N$. Let $\mu(\mathbf{x})$ and $\Sigma(\mathbf{x})$ be the empirical mean and variance of the sample of probability vectors $\mathbf{p}_1, \ldots, \mathbf{p}_N$. We can then minimize the trace of $\Sigma(\mathbf{x})$ to reduce variance.

The complete adversarial loss is given by

$$\mathcal{L}^{\star}_{\text{Art}} = \text{cross-entropy}(h(\mathbf{x}'), y_t) - \phi(\mathbf{x}') + \text{tr}(\Sigma(\mathbf{x}')).$$