[Reviews · NeurIPS 2019]

Reviewer 1



I like the paper, I think it is mostly nicely written, well motivated and the method seems solid. However there are some issues with the evaluation. While it seems that the authors did work hard to produce good evaluations, it is very hard to evaluate adversarial defense and I feel like more is needed for me to be convinced of the merits of these method. Detailed remarks: - The authors claim to use BPDA that is a method to overcome non-differentiable transformations, but there is nothing non-differentiable in objective 3 or 4 (unlike the number of steps). You can backpropagate through the gradients and it is commonly used in second order methods. These results should be added as well. - As is easily seen, in this work and others, there can be great variance in robustness against various attacks. To show validity of a defense it needs to be tested vs more attacks (and attacks targeted to fool the detector). I would be much more convinced if they showed robustness against different strong attacks, like the boundary attack that can be modified easily to work against detection. - While the results are ok, they are still bellow performance of robust networks like Madry et al so I am unsure about the practical significance (I know this is about detection not robust classification so the comparison isn't exact but still). - The main idea is combining two known properties of NN for detection so this work has slightly limited novelty.

Reviewer 2



This paper addresses the problem of detecting adversarial attacks to image classifiers -- this is a very important problem, and its solution lies at the heart of what can be considered to be one of the main challenges to overcome in the near future for this kind of classifiers to be of use in real-world applications. The solution proposed is novel in that the existence of so-called adversarial perturbations -- usually considered to be the main problem -- are used as the main building blocks for a defense mechanism against attacks. The material is clearly presented, and experimental results show that the approach achieves good results on a challenging dataset. The main question that comes up, on which the authors should comment in the rebuttal, is regarding one of the basic tenets on which the method is based. In particular, it's not clear whether the criterion regarding closeness to the decision boundary is universal -- is it valid for any kind of classifier? A discussion of limitations of the approach with respect to this issue seems necessary. Another question is regarding the white-box adversary: what prevents the design of an attack method using the authors' method as a black box, basically iterating in the generation of adversarial images until it passes the detection method? Finally, in the conclusion the authors mention in passing that running time is a limitation; could you please provide some details regarding this? As it is, it's not possible to determine if it's a major issue or just a minor drawback.

Reviewer 3



Though the proposed method has achieved improvements on standard datasets. However, there are some questions about the methodology. Why data close to the decision boundaries are considered adversarial examples? Though the densities of data near the decision boundaries are relatively sparse, should it be the core principle for finding adversarial examples? The writing quality of the paper is questionable. There are lots of grammar issues that may prevent readers from reading the paper smoothly. For example, line 8: "an image tampered with", what's the object following with? And, line 35, there should be a comma after "on one hand"

[Author Response · NeurIPS 2019]

We greatly appreciate that both R1 and R2 consider our paper to be well-written/clearly presented. We thank R3 for
pointing out spurious typos and will address them in the final version.

## Reviewer 1

**BPDA:** We investigated backpropagation through gradients on ImageNet and report
results in the table to the right. The detection rates are similar to those of using BPDA
(Table 1, combined) in the main text except that CW attack at LR = 0.01 becomes
stronger. The attack time is significantly longer: on average 31 sec/image compared
to 14 sec/image by BPDA.

| LR | FPR | PGD | CW |
|------|-----|-------|-------|
| 0.01 | 0.2 | 0.536 | 0.587 |
| 0.03 | 0.2 | 0.652 | 0.794 |
| 0.10 | 0.2 | 0.710 | 0.817 |
| 0.01 | 0.1 | 0.288 | 0.337 |
| 0.03 | 0.1 | 0.411 | 0.593 |
| 0.10 | 0.1 | 0.493 | 0.692 |

**Other attacks:** We note that PGD and CW are popular and fairly standard base
attacks that can be modified to produce strong white-box attacks against numerous
defenses as shown in [1, 5]. We further experimented with boundary attack for
attacking the model and detection mechanism as a black box, as suggested. Boundary attack is known to be very
sensitive to changes in the input, hence we omitted C2t/u in this evaluation in favor of speed but note that the detection
rate only worsens with this omission. On CIFAR-10, the boundary attack achieves a final average MSE of 0.009 (which
is comparable to the $L_\infty$ norm bound of 0.1 in other experiments), and our detector has a detection rate of $87.1\%$ at
FPR = 0.1, which is much higher than what we've obtained on white-box attacks (Table 2). We will include a detailed
evaluation in the final version.
**Madry et al.:** We note that our results (on CIFAR) are based on an $L_\infty$ perturbation norm bound of 0.1, which is much
larger than the bound of 0.03 used in Madry's work and makes detection much harder. To compare against their work,
we examined the difference between the accuracy on undefended ($95.3\%$) and adversarially trained models ($87.3\%$),
which is close to the $10\%$ FPR setting we used in our experiments. Thus, we further evaluated our detector using the
threshold of 0.03. Under the strongest PGD attack, our approach has a detection rate of $84.0\%$ (50-step PGD, LR = 0.1)
while Madry's adversarial training method has a recognition accuracy of $45.8\%$ (20-step PGD).
We would like to emphasize that our focus is on defending models trained on the more practical large-scale and diverse
ImageNet dataset, which few works have experimented or succeeded on (including Madry's, which only evaluates on
MNIST and CIFAR). We certainly do not claim that the detection rates obtained by our detector are sufficiently high,
but wish to inform the community of a previously unexplored technique of exploiting inherent trade-offs in strong,
adaptive white-box attacks. We hope that this aspect of our study can be appreciated.
**Known properties and novelty:** While the two properties are indeed known, our observations and analysis (cf. Sect.
4) that adversarial examples cannot obtain those properties simultaneously, even with adaptive attacks, are by no means
trivial. In particular, we consider our use of the apparent weakness of CNNs to adversarial examples as a strength to be
highly novel. The failure of existing ensemble defenses stems from the non-exclusivity of ensemble components, and
we are the first to show that exclusivity of detection criteria may be the solution to this difficult problem.

## Reviewer 2

**Closeness to decision boundaries:** We would like to clarify the fact that natural images in high-dimensional space are
close to decision boundaries is the underpinning of the existence of adversarial examples. Several works have attempted
to also theoretically prove that this is inevitable for *any* classifier (Lines 29-31). On the other hand, we empirically
show that when using CNN classifiers, adversarial examples will be far away from the boundary if they are optimized
with gradient descent to be robust to random noise. We admit that this empirical evidence, although convincing, is not a
proof that counterexamples do not exist. However, our experiments using PGD and CW modified to fool our detector
show that the existing framework for white-box attacks may be insufficient and more advanced techniques are required
to fully bypass our detection mechanism.
**Our method as a black box:** Please see our response to R1 for additional experiments using boundary attack.
**Detection time:** This is indeed an important factor to discuss that we omitted
in the draft. We have performed timing analysis (per image on average) on
the various components of our detector and included the results in the table
on the right side. It can be seen that C2t/u consumed most of the detection
time due to counting the number of steps of gradient descent required to cross
the decision boundary. CW takes much longer to detect since optimizing the
margin loss moves the adversarial example much further into the decision
boundary, requiring significantly more steps for C2t/u.

|  |  | PGD | CW |
|---------|-----|--------|--------|
| CIFAR | C1 | 0.013s | 0.012s |
|  | C2t | 0.128s | 0.27s |
|  | C2u | 0.055s | 14.23s |
| ImageNet | C1 | 0.091s | 0.107s |
|  | C2t | 1.057s | 3.46s |
|  | C2u | 0.138s | 0.241s |

## Reviewer 3

**Methodology:** We believe there is a misunderstanding regarding the core principle of our approach. We postulate
that points *far away from* decision boundaries are unlikely to occur naturally and are likely created by an adversary —
the fact that all natural images are close to the decision boundary is the exact reason for the existence of adversarial
examples. The essence of our paper is that this property of natural images is difficult to satisfy when the adversarial
image is also required to be robust to random noise — another property of CNNs trained on natural images.

[Meta-Review · NeurIPS 2019]

This paper presents a new method for detecting adversarial attacks, based on an analysis of the robustness to Gaussian noise and the closeness to decision boundaries. Overall, a good paper addressing an important problem. The comparison to previous detection methods is reasonably extensive, while some discussion on the reverse cross-entropy learning-based method (https://papers.nips.cc/paper/7709-towards-robust-detection-of-adversarial-examples.pdf) would be valuable.